# The Association between Gestational Diabetes and the Microbiome: A Systematic Review and Meta-Analysis

**DOI:** 10.3390/microorganisms11071749

**Published:** 2023-07-04

**Authors:** Rita Almeida Teixeira, Cláudia Silva, António Carlos Ferreira, Diana Martins, Adelino Leite-Moreira, Isabel M. Miranda, António S. Barros

**Affiliations:** Cardiovascular R&D Centre, UnIC@RISE, Department of Surgery and Physiology, Faculty of Medicine of the University of Porto, Alameda Professor Hernani Monteiro, 4200-319 Porto, Portugal

**Keywords:** microbiome, microbiota, dysbiosis, pregnancy, gestational diabetes

## Abstract

Gestational diabetes, affecting about 10% of pregnancies, is characterized by impaired glucose regulation and can lead to complications for health of pregnant women and their offspring. The microbiota, the resident microbes within the body, have been linked to the development of several metabolic conditions. This systematic review with meta-analysis aims to summarize the evidence on the differences in microbiota composition in pregnant women with gestational diabetes and their offspring compared to healthy pregnancies. A thorough search was conducted in the PubMed, Scopus, and Web of Science databases, and data from 21 studies were analyzed utilizing 41 meta-analyses. In the gut microbiota, Bifidobacterium and Alistipes were found to be more abundant in healthy pregnancies, while Roseburia appears to be more abundant in gestational diabetes. The heterogeneity among study findings regarding the microbiota in the meconium is considerable. The placental microbiota exhibited almost no heterogeneity, with an increased abundance of Firmicutes in the gestational diabetes group and a higher abundance of Proteobacteria in the control. The role of the microbiota in gestational diabetes is reinforced by these findings, which additionally point to the potential of microbiome-targeted therapies. To completely comprehend the interactions between gestational diabetes and the microbiome, standardizing methodologies and further research is necessary.

## 1. Introduction

Gestational diabetes mellitus (GDM) is a complication of pregnancy characterized by the development of hyperglycemia during gestation, diagnosed in the second or third trimester of pregnancy in women who do not yet have diabetes [1]. Women are usually screened between the 24th and the 28th week of gestation. It is estimated to affect approximately 11% of all pregnancies in Europe [2] and 7–10% worldwide [3]. This condition has the potential to have a significant impact on both maternal and fetal health [2,4]. Women with GDM have an increased risk of developing type 2 diabetes after pregnancy and experiencing complications such as pre-eclampsia, cesarean delivery, preterm delivery, fetal macrosomia, shoulder dystocia, and neonatal hypoglycemia [5].

While the exact mechanisms underlying the development of GDM are not entirely understood, recent evidence suggests that the microbiome may play a relevant role [5]. It is broadly accepted that during pregnancy, there is an imbalance between inadequate insulin secretion and the placental secretion of diabetogenic hormones, such as estrogen, progesterone, leptin, cortisol, placental lactogen (PL), and placental growth hormone (GH). This imbalance can lead to a decrease in peripheral insulin sensitivity as the pregnancy progresses [3]. Consequently, β-cell dysfunction is exacerbated by insulin resistance. Reduced insulin-stimulated glucose uptake further contributes to hyperglycemia, overburdening the β-cells, which must produce additional insulin in response. The direct contribution of glucose to β-cell failure is described as glucotoxicity. Thus, once β-cell dysfunction begins, a vicious cycle of hyperglycemia, insulin resistance, and further β-cell dysfunction is set in motion. This unstable metabolic condition, characterized by hyperglycemia and insulin resistance, leads to the development of GDM. In pregnancies associated with GDM, there is a notable increase in adipocyte fatty acid binding protein (FABP) expression and a decrease in peroxisome proliferator-activated receptor gamma (PPARγ) expression, along with chronic inflammation due to defective insulin receptor substrate-1 (IRS-1) function and insulin receptor phosphorylation [6]. This is associated with a proinflammatory state in which inflammatory cytokines, such as interleukin-6 (IL-6) and tumor necrosis factor-alpha (TNF-alpha), are activated, and there is a downregulation of interleukin-4 (IL-4) and interleukin-10 (IL-10) [7,8].

The microbiota refers to the complex community of microorganisms that inhabit various sites within the body, including the gut, skin, placenta, mouth, genitourinary and respiratory tract [9]. These microorganisms form the human microbiota, composed chiefly of bacteria but including viruses, archaea, molds, yeasts, and protozoa [10], representing over 100 trillion cells [11,12]. The microbiota are influenced by the environment surrounding an individual and modulates several conditions. Some established paths include the gut–brain axis [13] and the impact of the gut microbiome on metabolic conditions, such as diabetes, obesity, and non-alcoholic liver disease [14,15].

Although it is not yet fully understood exactly what contributions the microbiome may make to GDM pathogenesis, several studies report alterations in the microbiome composition found in patients with this complication and their offspring. In the context of gestational diabetes, the microbiome is thought to influence the development of insulin resistance, inflammation, and other metabolic disturbances characteristic of the disease. For example, dysbiosis in pregnancy alters short-chain fatty acid (SCFA) levels [16,17], the same mechanism thought to contribute to the other metabolic disease cited above. As such, several studies use microbiome modulation through probiotics to try to diminish the incidence and complications of GDM [18,19,20].

This systematic review with meta-analysis aims to update the previous reviews on this topic [21,22], including the new data available, summarizing and quantifying all relevant information to describe the current state of knowledge regarding the relationship between the microbiome and GDM. This review provides a comprehensive summary of the evidence on this topic, identifies areas of agreement and disagreement in the literature, and suggests directions for future research. The authors conducted a comprehensive literature search to identify relevant studies and used meta-analytic techniques to synthesize these studies’ findings and assess the evidence’s overall ability to achieve these goals. The objective of this systematic review with meta-analysis was to summarize the existing evidence on the differences in microbiota composition in pregnant women with gestational diabetes compared to healthy pregnancies.

## 2. Materials and Methods

This review was conducted according to the Preferred Reporting Items for Systematic Reviews and Meta-Analyses (PRISMA) statement, whose compliance checklist can be found in Appendix A [23].

### 2.1. Literature Retrieval

A comprehensive search of the literature was conducted on 13 November 2022, aiming to identify relevant studies on the relationship between the microbiome and gestational diabetes mellitus. The search was conducted in PubMed, Scopus, and Web of Science databases. The search was not limited to studies published in a specific language or time period. The QUERY utilized was (Microbiome [MeSH] AND gestational diabetes [MeSH]) OR (Microbiota [MeSH] AND gestational diabetes [MeSH]). The reference lists of identified studies and relevant review articles were also analyzed for additional studies.

As inclusion criteria were defined, studies presented data on the relative abundance of the microbiome in both gestational diabetes and control groups. Conversely, it was defined as exclusion criteria if: (1) the studies were reviews or systematic reviews; (2) comprised subjects that were not humans, such as animals or cells; (3) the study did not have a control group without gestational diabetes; (4) contained incomplete data, such as standard deviation or interval range; or (5) did not use comprehensive methods. 

The authors were contacted when there was no article access or the required information was unavailable. This analysis included all replies until the 27 February 2023.

### 2.2. Literature Screening

All the articles were screened by four researchers independently and blindly. The studies were selected based on titles and abstracts in the first phase. The full text was analyzed second, selecting the relevant articles. Studies that led to inconsistencies in the decision were reanalyzed by all researchers, who provided their reasons for inclusion or exclusion until a consensus was reached. Studies referring to maternal gut microbiota and mycobiota, fetal gut microbiota, meconium’s microbiota, placental microbiota, vaginal microbiota, maternal oral microbiota, fetal oral microbiota, and colostrum microbiota were included. 

### 2.3. Data Extraction

Using a self-developed data-extraction table to extract data, the articles were screened to retrieve the following information on the study characteristics: (1) title of the article, authors, publication year, and journal; (2) site of microbiome collection; (3) sample size; (4) study population; (5) method used to identify microbiota; (6) gene sequencing region analyzed; (7) moment when the sample was collected; and (8) taxonomic level of the results presented (e.g., phylum, genus). 

In a second phase, the same process was used to organize the information on the mean or median of relative abundances of each microbe in the gestational diabetes group and control and respective standard deviation or interval range. The information on this phase was obtained either directly from the article or its Appendix A by contacting the corresponding author. When the information was only available in an image, the underlying numerical data were extracted using WebPlotDigitizer: Version 4.6 software [24]. Three researchers measured the figures independently, and the results were compared to guarantee accuracy. 

### 2.4. Statistical Methods

RevMan 5.4.1 [25] was used to analyze the data extracted by elaborating meta-analysis and their respective forest plots. Meta-analysis was performed using the inverse variance with a random-effects model to calculate the standardized mean differences with 95% confidence intervals (CI). A random-effects model was used to estimate the pooled effect size due to the expected heterogeneity among studies. The heterogeneity among studies was evaluated by tau² and I² statistics. A *p*-value < 0.05 was considered statistically significant. A Z-test was used to test for overall effect size. A *p*-value < 0.05 was considered statistically significant.

Body sites without enough quantitative data to enable meta-analysis were not further considered. The meta-analysis was limited to articles that reported values for standard deviation or other statistical measures that could be converted to standardized effect sizes. Articles that did not meet this criterion were excluded from the analysis to ensure the accuracy and validity of the meta-analytic findings. Forest plots were extracted after analysis, as well as funnel plots. 

### 2.5. Quality Assessment and Risk of Bias

The individual quality and risk of bias of each study included were evaluated using study quality assessment tools of the National Heart, Lung, and Blood Institute of the National Institutes of Health (NIH) for quality assessment of observational cohort and cross-sectional studies and of case-control studies [26]. The studies’ quality was accessed by three investigators independently. The results were compared, and inconsistencies were debated until a consensus was reached. This information can be found in Appendix A.

## 3. Results

### 3.1. Study Selection

A total of 350 articles were obtained through a literature search. After the exclusion of 70 duplicates, 280 records were screened by title and abstract by four researchers individually. Of these, 49 were excluded because they were not original articles, 20 were excluded because they did not study humans, and 46 were excluded because they did not address the study’s research question. 

A citation tracking of the relevant articles was also undertaken, adding 66 studies which, after duplicate removal, resulted in 43 studies. Twenty-eight articles from this group already present in the original search were also removed. One of the articles could not be accessed, nor was an answer provided upon request. Thus, to the original 165 articles, 14 studies were added for the full-text analysis phase, resulting in 177 articles. 

Of the remaining articles, 21 were included in the meta-analysis. The reasons for exclusion of the remaining 156 articles were: 29 did not present relevant data, 19 did not have enough similar studies to enable the meta-analysis, 24 were not original articles, 6 studied an incorrect population (one studied only obese women, four did not have a control group, and one studied only women 5 years postpartum), 70 did not address the research question, 3 did not study humans, and 5 used only PCR techniques to measure relative abundance.

The reviewers disagreed with five articles during the screening process. All researchers reanalyzed these articles and provided their reasons for inclusion or exclusion until a consensus was reached. Cohen’s kappa coefficient was calculated and found to be 0.6828. The PRISMA [23] flow chart in Figure 1 summarizes the entire process.

### 3.2. Studies Characterization

As described in the previous section, 21 studies were included for meta-analysis. The studies were mainly conducted in China (*n*= 15), but there were also studies from the United States (*n* = 2), Mexico (*n* = 1), Japan (*n* = 1), Denmark (*n*= 1), and Italy (*n* = 1). The microbiota were collected from various sites, including the gut (*n* = 14), meconium (*n* = 4), and placenta (*n* = 3). Most studies used 16S rRNA gene sequencing to identify the microbiota (*n* = 20), with the remaining study using metagenome shotgun sequencing. Table 1 summarizes the studies’ characteristics.

### 3.3. Taxa Relative Abundances

After data collection, meta-analysis, and the respective forest and funnel plots were elaborated (Appendix A). The summarized results on the standard mean differences of relative abundances between the control group and gestational diabetes group can be found in Figure 2. For a comprehensive analysis and access to the raw data, refer to Appendix A.

#### 3.3.1. Gut Microbiota

The gut microbiota studies were the most abundant in our meta-analysis (*n* = 14). However, high heterogeneity was observed among studies, which required excluding some outlier studies to reduce the risk of bias. All the meta-analyses, both before and after adjustment, can be found in Appendix A.

Regarding the different phyla, Actinobacteria showed very high heterogeneity amongst studies—99% initially and 58% after removing two outlier studies. After removing the aforementioned studies, there was no statistically significant difference between the groups. Bacteroidetes were also highly heterogeneous among studies—99% initially—reducing to 33% after removing two studies, but no statistically significant results were found. Firmicutes showed similarly high heterogeneity (99% initially and 66% after the removal of a study) with no statistically significant results. Only three studies presented quantitative information regarding Fusobacteriota, which exhibited a heterogeneity of 75% among them. However, the results were not statistically significant. Proteobacteria also showed high heterogeneity (99% initially), but after the removal of the different study, it decreased to 0%. No statistically significant differences between the groups were observed. Verrucomicrobia also showed high heterogeneity (99% initially, becoming 68% after removing an outlier). This translated into a higher relative abundance of this phylum in the control group, with a standardized mean difference of −0.51, CI 95% [−0.87,−0.15].

Regarding genera, *Faecalibacterium* exhibited a heterogeneity of 23%, showing no statistically significant differences. Although *Prevotella* exhibited a 0% heterogeneity, no statistically significant differences were found between the groups. *Bacteroides* also exhibited a 0% heterogeneity, with no statistically significant differences between the groups. Even though *Megamonas* initially showed a heterogeneity of 99%, the removal of an outlier corrected it to 0%. However, there was no statistically significant difference between the groups. *Bifidobacterium* initially exhibited a heterogeneity of 71%, but it became 0% after removing one study. This showed a higher relative abundance in the control group than in the GDM group, with a standardized mean difference of −0.31, CI 95% [−0.50, −0.11]. *Escherichia and Shigella* exhibited a heterogeneity of 98% without any statistically significant results. *Roseburia* initially exhibited a heterogeneity of 68%, but it became 0% after excluding one study. This showed a higher relative abundance in the GDM group, with a standardized mean difference of 0.29, CI 95% [0.08, 0.51]. *Ruminococcus* exhibited a 0% heterogeneity amongst studies and no statistically significant differences between the groups. *Blautia* initially exhibited a heterogeneity of 93%, with a slight decrease to 83% after the removal of one study. *Alistipes* exhibited 0% heterogeneity among studies and were observed to have a higher abundance in the control group, with a standardized mean difference of −0.28, CI 95% [−0.53, −0.02]. 

*Parabacteroides*, *Coprococcus*, and *Clostridium* genera showed heterogeneities of 34%, 49%, and 28%, respectively. However, none of these yielded statistically significant differences between the groups.

#### 3.3.2. Meconium Microbiota

Concerning meconium’s microbiota, the meta-analysis showed that the Acidobacteria phylum had a heterogeneity of 0%, and no statistically significant differences were found in the relative abundances between the groups.

Similarly, the Actinobacteria phylum had a heterogeneity of 35%, but no statistically significant differences were observed between the groups. The meconium microbiota analysis revealed that the Bacteroidetes, Cyanobacteria, Firmicutes, and Proteobacteria phyla exhibited a heterogeneity of 74%, 59%, 44%, and 65%, respectively, with no statistically significant differences between the GDM and control groups.

Concerning the analysis of genera, *Acinetobacter* had a heterogeneity of 31% between studies and presented a non-statistically significant increase in relative abundance in the GDM group. On the other hand, *Staphylococcus* showed a statistically significant increase in relative abundance in the GDM group, despite an extremely high heterogeneity of 99% amongst studies. Furthermore, the genera *Pseudomonas*, *Paracoccus*, and *Enhydrobacter* also had a heterogeneity of 99%, with a statistically significant higher relative abundance in the control group. Finally, *Serratia* presented a heterogeneity of 55% among studies without statistically significant differences between the groups.

Overall, the meta-analysis found no statistically significant differences in the relative abundances of meconium microbiota phyla and genera between the GDM and control groups.

#### 3.3.3. Placental Microbiota

Despite the reduced amount of articles studying the placental microbiome, unlike other body sites, the studies that presented data on placental microbiota had a low heterogeneity among them. 

The meta-analysis found no significant differences in the relative abundance of Actinobacteria or Bacteroidetes phyla between patients with GDM and controls. However, a significant increase in the relative abundance of Firmicutes was observed in the GDM group (standard mean difference of 0.56, CI 95% [0.20, 0.92]). A significant increase in the relative abundance of Proteobacteria was observed in the control group compared to the GDM group (standard mean difference of −0.34, CI 95% [−0.62, −0.05]).

Regarding genera, a significant increase in the relative abundance of *Faecalibacterium* was observed in the GDM group (standard mean difference of 0.44, CI 95% [0.05, 0.83]), as well as in *Bacteroides* and *Lachnospiraceae* uncl. (standard mean differences of 0.40, CI 95% [0.11, 0.70] and 0.32, CI 95% [0.03, 0.61], respectively). No significant differences were observed in the *Blautia* and *Bifidobacterium* genera.

## 4. Discussion

### 4.1. Microbiome

#### 4.1.1. Gut Microbiome

Most of the bacteria in the meta-analysis did not present differences in the relative abundance between groups. 

Classically, Bacteroidetes and Firmicutes represent 90% of the gut’s microbiota [48], which could mostly be confirmed by the relative abundance of data collected throughout the analyzed studies. Moreover, the Firmicutes/Bacteroidetes ratio is often used as a marker of dysbiosis [49]. Despite this, it was not possible to find statistically significant differences in both phyla between the GDM and the control group. 

After removing the most heterogeneous study, the heterogeneity of Verrucomicrobia was reduced, revealing a higher relative abundance in the control group. This finding may have metabolic implications, given that Verrucomicrobia has been associated with the production of short-chain fatty acids (SCFAs) that improve glucose tolerance and insulin sensitivity. Moreover, Verrucomicrobia is known to form a close symbiotic community of cells that line the inner surface of the intestine and reinforce the intestinal barrier by reducing its permeability. These properties suggest that an increased abundance of Verrucomicrobia in the control group may benefit metabolic health [50,51]. Despite the logical coherence of these findings with the hypothesized pathology mechanism, it is nonetheless worth considering the high heterogeneity among the studies, which may constitute a limitation to the validity of this finding. 

Similarly, the higher relative abundance of *Bifidobacterium* in the control group found in our analysis is consistent with previous studies reporting the beneficial role of *Bifidobacterium* in glucose homeostasis and inflammation. *Bifidobacterium* is known as a beneficial acetate- and lactate-producing bacteria. The diminished relative abundance of this genus in the GDM group can be interpreted considering the role of *Bifidobacteria* in the fermentative metabolism, carbohydrate degradation, and intracellular uptake of short oligosaccharides [22,52,53]. These findings are consistent with the recommendation of probiotic supplementation based on this genus [54].

The *Blautia* genus was pointed to in the literature as a marker of dysbiosis in disease situations. Nonetheless, the studies’ analysis shows such a high level of heterogeneity that no conclusion can be made relative to the higher prevalence of this genus in one group or the other. 

*Alistipes* exhibited no heterogeneity among studies and was observed to have a higher abundance in the control group, which is coherent with the thought that this genus is a protective ally in cardiovascular diseases [55].

Conversely, *Roseburia’s* heterogeneity of 0% after the exclusion of one study leads to the understanding that there is a higher relative abundance of this genus in the GDM group. This contradicts the belief that an increased abundance of *Roseburia*, a butyrate-producing anaerobic bacteria, is associated with weight loss and reduced glucose intolerance [56], being a marker of health [57].

#### 4.1.2. Meconium Microbiome

Although a great number of studies have analyzed the changes in the relative abundances of meconium microbiota, only a small portion provided the quantitative data to allow their introduction into the meta-analysis. 

The meta-analyses examining the association between meconium microbiota and GDM demonstrated significant heterogeneity among studies. Notably, the results were not statistically significant for most bacterial genera and phyla assessed. 

These findings suggest that further research is necessary to clarify the potential alterations of meconium microbiota in the offspring of women with gestational diabetes. It is important to note that several factors, including differences in study design, sample collection and processing methods, and analytical techniques, may influence the observed heterogeneity [58]. As such, careful consideration of these factors is warranted when interpreting the meta-analysis results and designing future studies.

#### 4.1.3. Placental Microbiome

Until recently, the placenta was considered sterile tissue. While recent studies have suggested the existence of a placental microbiome [59]—with some similarities to the oral microbiota [60]—some still consider it to be the result of contamination rather than true colonization [61]. The scientific community has been raising questions about its potential impact on pregnancy outcomes [60,62,63]. 

Despite the limited number of studies available for inclusion in the meta-analysis (*n* = 3), the results indicated very low heterogeneity between studies, suggesting that a dysbiotic placental microbiome may be associated with GDM. The findings also suggested that the observed placental microbiota is likely to be a true reflection of colonization rather than contamination. 

On the one hand, the increase in Firmicutes in the GDM group is aligned with the idea that Firmicutes may indicate a state of dysbiosis. On the other hand, the increased abundance of Proteobacteria in the control group contradicts the classical idea of Proteobacteria being a phylum associated with a disease state, often seen with increased inflammation [64]. 

Regarding genera, a significant increase in the relative abundance of *Faecalibacterium*, a butyrate-producing genus considered a healthy bacteria, was observed with a higher relative abundance in the GDM group, contrary to what was expected [65].

*Bacteroides* are presented with a higher relative abundance in the GDM group, which is consistent with the literature stating that this genus in the placenta is associated with more inflammation and dysbiosis [66,67]. 

No significant differences were observed in *Blautia* in the placental microbiome of the group with gestational diabetes and the control. However, literature reports that *Blautia’s* relative abundance in the gut’s microbiota had a strong inverse relationship between levels of Hba1c and insulin resistance [68]. 

These findings provide important insights into the potential role of the placental microbiome in pregnancy outcomes and highlight the need for further research to better understand the underlying mechanisms of these associations. Moreover, the consistency of the findings across the included studies strengthens the evidence base supporting the existence of a placental microbiome.

### 4.2. Considerations

It is important to highlight that these findings are based on observational studies. Therefore, it is not possible to establish a causal or consequential relationship between microbiota and metabolic outcomes in women with GDM. 

While the average relative abundance is a common and useful used metric in microbiota studies, it can, however, preclude some insights. Using the relative average abundance can: (1) hinder the inter-individual variations; (2) mask the dynamic variations of the microbial communities that may be related to important health-related outcomes; and (3) discard rare taxa that may have clinical importance. Indeed, longitudinal studies may provide a better understanding of the dynamic nature of the microbiota, enabling the identification of temporal patterns as a function of, for instance, interventions or changes in lifestyle.

In this study, a deliberate decision was made to exclude studies that used PCR techniques or other methods to search for specific bacteria in the characterization of the microbiota profile. This was justified by the belief that targeted approaches could introduce bias in estimating relative microbial abundances and, consequently, have a negative impact on the study results. Specifically, these targeted methods rely on primers or probes designed to amplify or detect specific bacterial taxa, which may not accurately represent the full microbial diversity in the sample. This is due to the fact that different articles can focus on different targets, leading to different results.

Instead, it was preferred to focus on articles that used untargeted techniques, such as 16S rRNA sequencing or metagenomic shotgun sequencing, which can provide a more comprehensive microbiota profile. These methods provide a more holistic view of the microbial communities in the samples. They do not rely on predefined targets, offering the possibility to capture the diversity and abundance of all microbial taxa, including those that may not have been previously characterized.

Since for most of the body sites analyzed, there was a sufficient number of studies using the desired techniques, it was considered that this selection would not affect the ability to summarize the actual evidence. While this decision may have led to the exclusion of some potentially relevant studies, it was thought necessary to ensure our findings’ robustness and generalizability. By focusing on articles that used more comprehensive methods, a more accurate and representative picture of the patient’s microbiota was considered. Thus, this approach anticipated problems with introducing heterogeneity among studies due to the similarity of methodology and identified trends that may be more broadly applicable across populations. This approach also has the limitation of relying on databases that are still under improvement. 

It is worth noting that this decision has its limitations. Even with broader methods, biases may still be introduced by differences in sampling or analysis techniques across studies [69]. However, this approach was the most appropriate for the goals of our review and provides a solid foundation for further research in this area. 

Finally, the lack of quantitative data also led to occasional assumptions being made from figures presented instead of values provided by authors. Although this process is considered accurate, some values could perhaps differ from the original ones. 

## 5. Conclusions

Despite a significant improvement in the knowledge regarding the association of gestational diabetes mellitus to the microbiota of different body sites, further characterization of this association is still needed. One of the main challenges encountered while synthesizing the evidence was the limited availability of quantitative data and associated effect measures across the included studies. As a result, the number of eligible studies was reduced by more than half when presenting quantitative evidence to support the reported findings.

Furthermore, considerable heterogeneity was observed throughout the meta-analysis, underscoring the potential benefits of adhering to international protocols to standardize the findings and facilitate their interpretation and comparison [70,71]. Overall, while the present review provides valuable insights into the association between GDM and the microbiota of different body sites, future research should focus on generating and sharing more robust quantitative data with standardizing methodologies to enhance the comparability of results across studies [72,73]. An association would be established, allowing advantages to be taken from a clinical actuation perspective to minimize effects on the cause or consequences of this relation.

## Figures and Tables

**Figure 1 microorganisms-11-01749-f001:**
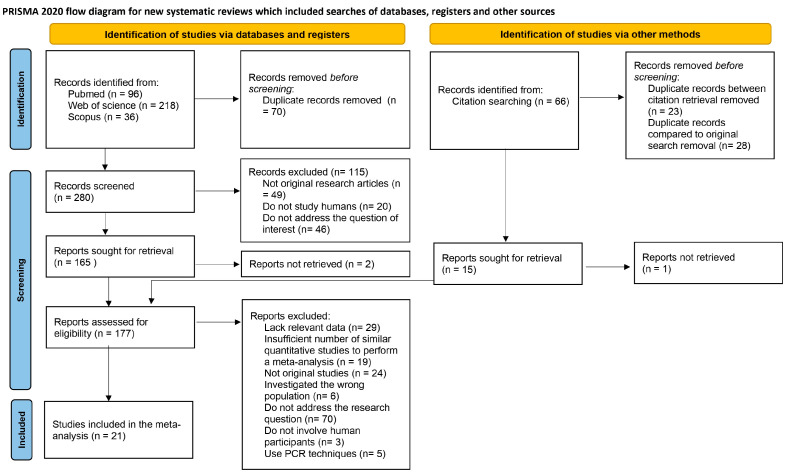
PRISMA flow chart [23].

**Figure 2 microorganisms-11-01749-f002:**
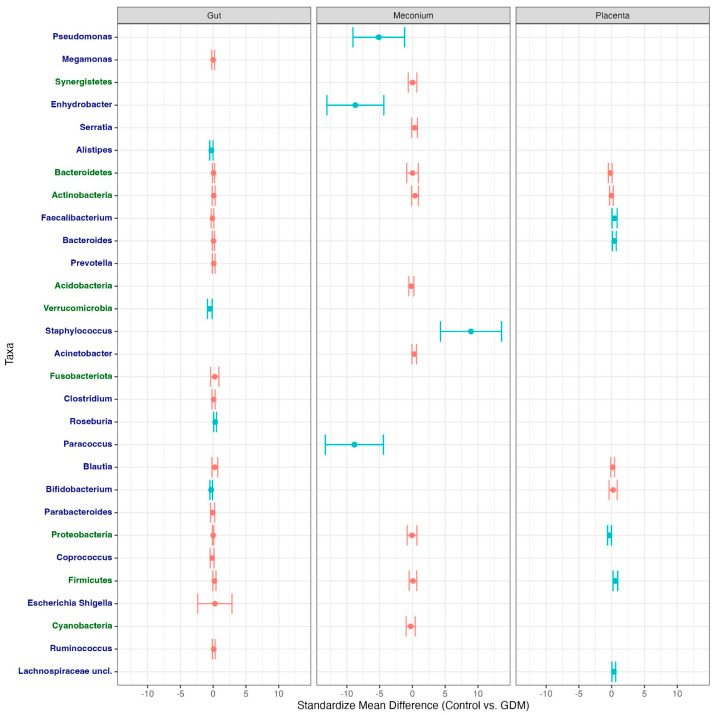
Standard mean differences of mean relative abundances of different taxa (phylum in green and genus in blue) between the control group and the gestational diabetes mellitus group. Represented in red are intervals with no statistically significant differences, and in aqua are the ones with statistically significant differences for IC 95%. Negative values mean higher relative abundance in the control group, and positive values mean higher relative abundance in the group with gestational diabetes.

**Table 1 microorganisms-11-01749-t001:** Study characterization.

Study		Body Part			N			Sequencing			
Author	year	G ^1^	M ^2^	P ^3^	Total	Diabetes	Control	Method	Region	Cohort Country	Sample Collection
Bassols, J. et al. [27]	2016			x	207	50	157	16S rRNA	V1–V2	Denmark	3rd trimester
Benítez-Guerrero, T., et al. [28]	2022	x			41	11	30	16S rRNA	V3	Mexico	3rd trimester
Hu, J., et al. [29]	2013		x		18	5	13	16S rRNA	V3–V4	USA	Meconium
Hu, P., et al. [30]	2021	x			402	201	201	16S rRNA	V3–V4	China	6–15 weeks
Su, M., et al. [31]	2018		x		34	20	14	16S rRNA	V4	China	Meconium
Su, Yao, [32]	2021	x			122	34	88	16S rRNA	V3–V4	China	24–28 weeks
Tanaka, et al. [33]	2022	x			36	20	16	16S rRNA	V3–V4	Japan	At the time of GDM diagnosis (T1) and at 35–37 weeks of gestation (T2)
Tang, N., et al. [34]	2020			x	15	8	7	16S rRNA	V3–V4	China	Placenta
Wang, J., et al. [35]	2018	x			147	74	73	16S rRNA	V3–V4	China	1–2 days postpartum, meconium
Wang, X., et al. [36]	2020	x			107	59	48	16S rRNA	V3–V4	China	24–28 weeks
Wei, J., et al. [37]	2022	x			33	15	18	16S rRNA	V3–V4	China	24–28 weeks
Wu, N., et al. [38]	2021	x			57	27	30	16S rRNA	V4	China	24–28 weeks
Wu, Y., et al. [39]	2020	x			49	23	26	metagenome shotgun	—	China	3rd trimester
Xu, Y., et al. [40]	2020	x			124	43	81	16S rRNA	V3–V4	China	21–29 weeks
Ye, G., et al. [41]	2019	x			52	36	16	16S rRNA	V3–V4	China	24–28 weeks
Zhang, X., et al. [42]	2021	x			34	7	27	16S rRNA	V4	China	2nd trimester
Zhang, Y., et al. [43]	2021	x			837	128	709	16S rRNA	V3–V4	China	22–24 weeks
Zheng, W., et al. [44]	2020	x			134	31	103	16S rRNA	V3–V4	China	2nd trimester
Zhu, Q., et al. [45]	2022		x		120	60	60	16S rRNA	V3	China	Meconium
Guzzardi, Maria Angela, et al. [46]	2022		x		79	21	58	16S rRNA	V3–V4	Italy	Meconium and childhood
Olomu IN, et al. [47]	2020			x	20	10	10	16S rRNA	V3–V4	USA	Placenta

^1^ G—Gut microbiota, ^2^ M—Meconium microbiota, ^3^ P—Placenta microbiota.

## Data Availability

The data presented in this study are available in the Appendix A.

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
