# Peer review of "The Association between Gestational Diabetes and the Microbiome: A Systematic Review and Meta-Analysis"

_microorganisms, 2023, doi:10.3390/microorganisms11071749_

Round 1
Reviewer 1 Report
The systematic review entitled The association between gestational diabetes and the microbiome: a systematic review and meta-analysis, provides insight into microbiota compositions sampled from different locations from gestational diabetes cases. the review is well executed and the process is very clear. i have just some minor comments which are listed bellow.
L 163 - 165: should be leftover from the template.
Fig 2: I assume the red and aqua colors represent the control and diabetes groups or vice versa. this information is missing from the figure legend.
the author mainly presented an abundance of microbiota across different sampling locations. however, I'm wondering if running a correlation study based on the extracted data is possible to see the correlation between microbiota and other factors.
Author Response
The systematic review entitled The association between gestational diabetes and the microbiome: a systematic review and meta-analysis, provides insight into microbiota compositions sampled from different locations from gestational diabetes cases. the review is well executed and the process is very clear. i have just some minor comments which are listed bellow.
Q1: L 163 - 165: should be leftover from the template
A1: Thank you for the suggestion. We agree that the Results paragraph preamble is unnecessary. We have removed from the manuscript.
Q2: Fig 2: I assume the red and aqua colors represent the control and diabetes groups or vice versa. this information is missing from the figure legend.
A2: Thank you for noting a potential confusion for the reader. Standard mean differences (SMD) of mean relative abundances between the control group and the gestational diabetes mellitus group of different taxa are depicted, in green for phylum, and blue for genus. Represented in red are intervals with no statistically significant differences, and in aqua are the ones with statistically significant differences for IC 95%. Negative values mean higher relative abundance in the control group, and positive values mean higher relative abundance in the group with gestational diabetes. Fig 2. caption is changed accordingly.
Q3: the author mainly presented an abundance of microbiota across different sampling locations. However, I’m wondering if running a correlation study based on the extracted data is possible to see the correlation between microbiota and other factors.
A3: Thank you for your suggestion. That would be a very interesting insight, however, unfortunately, not all considered works have all the necessary information to perform a Spearman correlation for detecting correlation patterns.

Reviewer 2 Report
I consider the manuscript well structured and presented. I have no further suggestions for the authors.
I consider the manuscript well structured and presented. I have no further suggestions for the authors.
Author Response
We thank the reviewer for taking the time to read the manuscript.
Reviewer 3 Report
Teixeira and colleagues did a nice systematic review and meta-analysis of the microbiome in gestational diabetes mellitus. Their search terms were rather narrow and only included MeSH terms, but this should still have retrieved most relevant articles.
The review only addresses changes in untransformed relative abundances. This is ok as far as study design choices go, but it is important to discuss limitations with this approach as well, since relative abundances are always confounded by all other microbes in the sample. It is also crucial to be very explicit with this design choice, since the introduction mostly speaks of "the microbiome" in abstract, but this can be operationalized in many different ways.
Further comments below:
lines 46/47: while mutations in IRS have been linked to GDM, there is no support to that hormonal imbalance in itself would lead to DNA-level mutations
lines 79+: define the goal more clearly. Microbiomes can be compared in any number of ways. Is it the case that this meta-analyses concerns itself only with relative abundance of specific taxa of interest? If so, justify this choice; if not, clarify the goal. This includes updating PRISMA point 4, "Objectives". "Relationship between microbiome and GDM" is too broad and encompasses a lot more than the work presented here.
line 123: Was any other information extracted, e.g. alpha-diversity indexes? If not, then why not?
lines 216-217: refer to suppl. fig 2 at the end of this paragraph
lines 277-279: this is in contradiction with fig. 2. If you believe fig. 2 is being driven by outliers and is unreliable, this should be clearly shown. I'd suggest putting the "raw" results with all the studies in the supplementary, and the "corrected" results which you actually trust in the main, to make the text and figures coherent. This also related to PRISMA 20a - Figure 2 does not make clear "the charachteristics and risks of bias among contributing studies"
lines 380-383: consider that the healthy placenta has very low biomass, and that Proteobacteria (such as E. coli) are very common laboratory contaminants. It is possible that an increase in Proteobacteria is simply reflecting lower biomass and therefore noisier data.
lines 409-416: while it is true that qPCR based studies will not be good indicators of relative abundance, they're a much better indicator of absolute abundance. Since in this paper you choose to study individual genera and phyla, not composite measures such as Chao1 richness or Shannon's entropy, it would be very interesting to see the qPCR results as well - however, they must be compared amongst themselves, since they're indeed not comparable to 16S-based studies.
line 142: that's a lot of criteria for heterogeneity. Is a significant p-value in any of them considered significant, in all of them, or how are they harmonized?
lines 172-176: I can't make these numbers add up. 44-28 is 16, and excluding one leaves 15, not 14. Also, 165+14 is 179, not 177.
Similar issues in fig. 1, 165+15 should be 180, but is somehow only 177.
line 42: is this a reference that wasn't included in the reference count?
line 123: these are taxonomic levels, not orders of magnitude.
lines 163-165: exclude
line 190: exclude "All"
Throughout the supplementary: correct typo "outlieres"
Author Response
Q1: Teixeira and colleagues did a nice systematic review and meta-analysis of the microbiome in gestational diabetes mellitus. Their search terms were rather narrow and only included MeSH terms, but this should still have retrieved most relevant articles.
A1: Thank you for your pertinent remark. In addition to this research, the references of the various articles were backtracked, so we believe that all papers relevant to the topic were included, with no other articles found in the other systematic reviews or meta-analyses that were not evaluated by the group.
Q2: The review only addresses changes in untransformed relative abundances. This is ok as far as study design choices go, but it is important to discuss limitations with this approach as well, since relative abundances are always confounded by all other microbes in the sample. It is also crucial to be very explicit with this design choice, since the introduction mostly speaks of "the microbiome" in abstract, but this can be operationalized in many different ways
A2: Thank you for you remark. It is true that relative abundances as its limitations, unfortunately the surveyed papers did not express the results in terms of absolute values (which will be the ideal case). Still, we have avoided any further transformation to due to the risk of increasing bias on the reported results. Following your suggestion, we will highlight this information in the paper. Regarding the second half of the question, we will also emphasize the concept of microbiota since microbiome and microbiota are high inter-dependent. The following paragraph was included in the manuscript:
Lines 420-427: “While the average relative abundance is a commonly and useful used metric in microbiota studies it can, however, preclude some insights. Using the relative average abundance can: 1) hinder the inter-individual variations; 2) mask the dynamic variations of the microbial communities that may be related to important health-related outcomes and 3) can discards rare taxa that may have clinical importance. Indeed, longitudinal studies may provide a better understanding of the dynamic nature of the microbiota, enabling the identification of temporal patterns as a function of, for instance, interventions or changes in lifestyle.”
Further comments below:
Q3: lines 46/47: while mutations in IRS have been linked to GDM, there is no support to that hormonal imbalance in itself would lead to DNA-level mutations
A3: Thank you for pointing out that the way the sentence is written may lead to scientific inaccuracies. Your feedback is greatly appreciated. Upon reviewing the paragraph concerning the pathophysiology of gestational diabetes, we have made revisions to ensure that the described pathophysiology is explicit:
Lines 46-60: “It is broadly accepted that during pregnancy, there is an imbalance between inadequate insulin secretion and the placental secretion of diabetogenic hormones, such as estrogen, progesterone, leptin, cortisol, placental lactogen (PL), and placental growth hormone (GH). This imbalance can lead to a decrease in peripheral insulin sensitivity as the pregnancy progresses [3]. Consequently, β-cell dysfunction is exacerbated by insulin resistance. Reduced insulin-stimulated glucose uptake further contributes to hyperglycemia, overburdening the β-cells, which must produce additional insulin in response. The direct contribution of glucose to β-cell failure is described as glucotoxicity. Thus, once β-cell dysfunction begins, a vicious cycle of hyperglycemia, insulin resistance, and further β-cell dysfunction is set in motion. This unstable metabolic condition, characterized by hyperglycemia and insulin resistance, leads to the development of GDM. In pregnancies associated with GDM, there is a notable increase in adipocyte fatty acid binding protein (FABP) expression and a decrease in peroxisome proliferator-activated receptor gamma (PPARγ) expression, along with chronic inflammation due to defective insulin receptor substrate-1 (IRS-1) function and insulin receptor phosphorylation [6]. This is associated with a proinflammatory state in which inflammatory cytokines, such as interleukin-6 (IL-6) and tumor necrosis factor-alpha (TNF-alpha), are activated, and there is a down-regulation of interleukin-4 (IL-4) and interleukin-10 (IL-10) [7,8].”
Q4: lines 79+: define the goal more clearly. Microbiomes can be compared in any number of ways. Is it the case that this meta-analyses concerns itself only with relative abundance of specific taxa of interest? If so, justify this choice; if not, clarify the goal. This includes updating PRISMA point 4, "Objectives". "Relationship between microbiome and GDM" is too broad and encompasses a lot more than the work presented here.
A4: Thank you for your comment. Your concerns are indeed connected to question Q2 and answer A2. We acknowledge that in our analysis, we focused only on taxa that appeared in certain number of studies (at least 3), allowing us to extract meaningful information. This limitation is a common challenge in the fields of microbiota and microbiome research due to the absence of: 1) standardized data harmonization practices; 2) clear guidelines for analysis workflows; and 3) effective incentives or mechanisms for sharing raw data. These issues, unfortunately, hamper comprehensive analysis and hinder progress in these fields. The following sentence was added to the introduction (lines 89-91) in order to make the objective more clear: “The objective of this systematic review with meta-analysis was to summarize the existing evidence on the differences in microbiota composition in pregnant women with gestational diabetes compared to healthy pregnancies.”
Changes were also updated in PRISMA point 4.
Q5: line 123: Was any other information extracted, e.g. alpha-diversity indexes? If not, then why not?
A5: True, indeed, it is important to collect alpha-diversity data. Regrettably, the availability of such information was limited across the papers under examination, once again preventing a meaningful analysis from being conducted.
Q6: lines 216-217: refer to suppl. fig 2 at the end of this paragraph
A6: correction made in the revised manuscript.
Q7: lines 277-279: this is in contradiction with fig. 2. If you believe fig. 2 is being driven by outliers and is unreliable, this should be clearly shown. I'd suggest putting the "raw" results with all the studies in the supplementary, and the "corrected" results which you actually trust in the main, to make the text and figures coherent. This also related to PRISMA 20a - Figure 2 does not make clear "the charachteristics and risks of bias among contributing studies".
A7: Thank you. Figure 2 illustrates the results of the conducted meta-analyses, excluding the outliers. For a comprehensive analysis and access to the raw data, please refer to Supplementary Material 2.
Additionally, PRISMA 20a answer was corrected to “Supplementary Material 2 - Meta-analysis and Supplementary Material 3 - Article Bias Evaluation - National Institute of Health”, which was considered a better match to what was required.
Q8: lines 380-383: consider that the healthy placenta has very low biomass, and that Proteobacteria (such as E. coli) are very common laboratory contaminants. It is possible that an increase in Proteobacteria is simply reflecting lower biomass and therefore noisier data.
A8: True, to draw accurate conclusions about the presence and significance of Proteobacteria in the healthy placenta, it is crucial to carefully account for the low biomass and consider potential contamination sources during sample collection and processing. Implementing rigorous controls and quality assurance measures can help address these concerns and ensure the reliability of the findings. Once more this highlights the lack of standardization guidelines on the reporting studies.
Q9: lines 409-416: while it is true that qPCR based studies will not be good indicators of relative abundance, they're a much better indicator of absolute abundance. Since in this paper you choose to study individual genera and phyla, not composite measures such as Chao1 richness or Shannon's entropy, it would be very interesting to see the qPCR results as well - however, they must be compared amongst themselves, since they're indeed not comparable to 16S-based studies.
A9: The reviewer raises an interesting point. However, based on our per-protocol approach for the pre-specified analysis of the manuscripts, we made the decision to exclude results obtained through PCR. In future studies within our group, we will certainly take that suggestion into account.
Q10: line 142: that's a lot of criteria for heterogeneity. Is a significant p-value in any of them considered significant, in all of them, or how are they harmonized?
A10: True, the sentence was confusing, thank you for the comment. We have changed in the manuscript and simplified the sentence to: “The heterogeneity among studies was evaluated by Tau², and I² Statistics. A P-value < 0.05 was considered statistically significant.” (lines 153-155).
Q11: lines 172-176: I can't make these numbers add up. 44-28 is 16, and excluding one leaves 15, not 14. Also, 165+14 is 179, not 177.
Similar issues in fig. 1, 165+15 should be 180, but is somehow only 177.
A11: Thank you for bringing up your concerns regarding the calculations in the manuscript. After carefully reviewing your comments, we have reevaluated the calculations and would like to provide a clarification.
The discrepancy arises from the fact that the articles which were not retrieved in both directions should have been subtracted before adding the remaining values. Thus, the correct calculation should be: (165 - 2) + (15 - 1) = 177.
Indeed, the correct subtraction of duplicate articles should have been 23 instead of 22. With this correction, the revised calculation now accurately accounts for the articles retrieved in both directions. Consequently, the updated numbers align correctly.
We have promptly rectified this mistake in the revised version of the manuscript, ensuring the accuracy of the calculations and maintaining the integrity of the findings. Your vigilance in identifying these errors has been invaluable in improving the quality of our work.
Q12: line 42: is this a reference that wasn't included in the reference count?
A12: Thank you for pinpointing that issue. The error (a parenthesis mistake) was corrected in the manuscript.
Q13: line 123: these are taxonomic levels, not orders of magnitude.
A13: correction made in the revised manuscript.
Q14: lines 163-165: exclude
A14: correction made in the revised manuscript.
Q15: line 190: exclude "All"
A15: correction made in the revised manuscript.
Q16: Throughout the supplementary: correct typo "outlieres"
A16: correction made all along the revised manuscript.
